# Dbx1 precursor cells are a source of inspiratory XII premotoneurons

**Ann L Revill[1], Nikolas C Vann[2], Victoria T Akins[2], Andrew Kottick[2], Paul A Gray[3], Christopher A Del Negro[2], Gregory D Funk[1]***

[1]Department of Physiology, Neuroscience and Mental Health Institute, Women and Children's Health Research Institute, Faculty of Medicine and Dentistry, University of Alberta, Edmonton, Canada; [2]Department of Applied Science, The College of William and Mary, Williamsburg, United States; [3]Department of Anatomy and Neurobiology, Washington University School of Medicine, St. Louis, United States

**Abstract** All behaviors require coordinated activation of motoneurons from central command and premotor networks. The genetic identities of premotoneurons providing behaviorally relevant excitation to any pool of respiratory motoneurons remain unknown. Recently, we established in vitro that Dbx1-derived pre-Bötzinger complex neurons are critical for rhythm generation and that a subpopulation serves a premotor function (*Wang et al., 2014*). Here, we further show that a subpopulation of Dbx1-derived intermediate reticular (IRt) neurons are rhythmically active during inspiration and project to the hypoglossal (XII) nucleus that contains motoneurons important for maintaining airway patency. Laser ablation of Dbx1 IRt neurons, 57% of which are glutamatergic, decreased ipsilateral inspiratory motor output without affecting frequency. We conclude that a subset of Dbx1 IRt neurons is a source of premotor excitatory drive, contributing to the inspiratory behavior of XII motoneurons, as well as a key component of the airway control network whose dysfunction contributes to sleep apnea.

*For correspondence: gf@ualberta.ca

**Competing interests:** The authors declare that no competing interests exist.

## Introduction

Understanding the genetic basis of behavior is a fundamental goal of neuroscience that requires functional identification of constituent neurons, knowledge of their embryonic origins, and characterization of their cellular, synaptic and modulatory properties (*Goulding, 2009*; *Garcia-Campmany et al., 2010*). Central pattern generator (CPG) circuits comprise rhythmogenic and pattern-forming components that underlie rhythmic behaviors including breathing, locomotion, and mastication. The efferent motor network controlling inspiratory activity of tongue protruder muscles maintains airway patency and is an integral part of breathing behavior. This network minimally comprises the rhythmogenic pre-Bötzinger Complex (preBötC), XII motoneurons that drive tongue protruder muscles, and inspiratory premotoneurons intercalated between the two (*Koizumi et al., 2008*; *Fregosi, 2011*) (*Figure 1A*).

Dbx1-derived preBötC neurons underlie rhythmogenesis and contribute some premotor function (*Wang et al., 2014*). However, XII premotoneurons are predominantly located in the intermediate medullary reticular formation (IRt) (*Ono et al., 1994*; *Dobbins and Feldman, 1995*; *Woch et al., 2000*; *Peever et al., 2002*; *Koizumi et al., 2008*; *Stanek et al., 2014*), a heterogeneous region that contributes to orofacial behaviors (*Gestreau et al., 2005*; *Moore et al., 2013*; *Kleinfeld et al., 2014*; *Moore et al., 2014*). The IRt contains Dbx1-derived neurons (*Gray, 2008*, *2013*; *Ruangkittisakul et al., 2014*) and diffusely distributed XII inspiratory premotoneurons (*Koizumi et al., 2008*). Because Dbx1 progenitors are a source of glutamatergic IRt neurons (*Gray, 2008*, *2013*; *Ruangkittisakul et al., 2014*) and inspiratory motor drive is glutamatergic

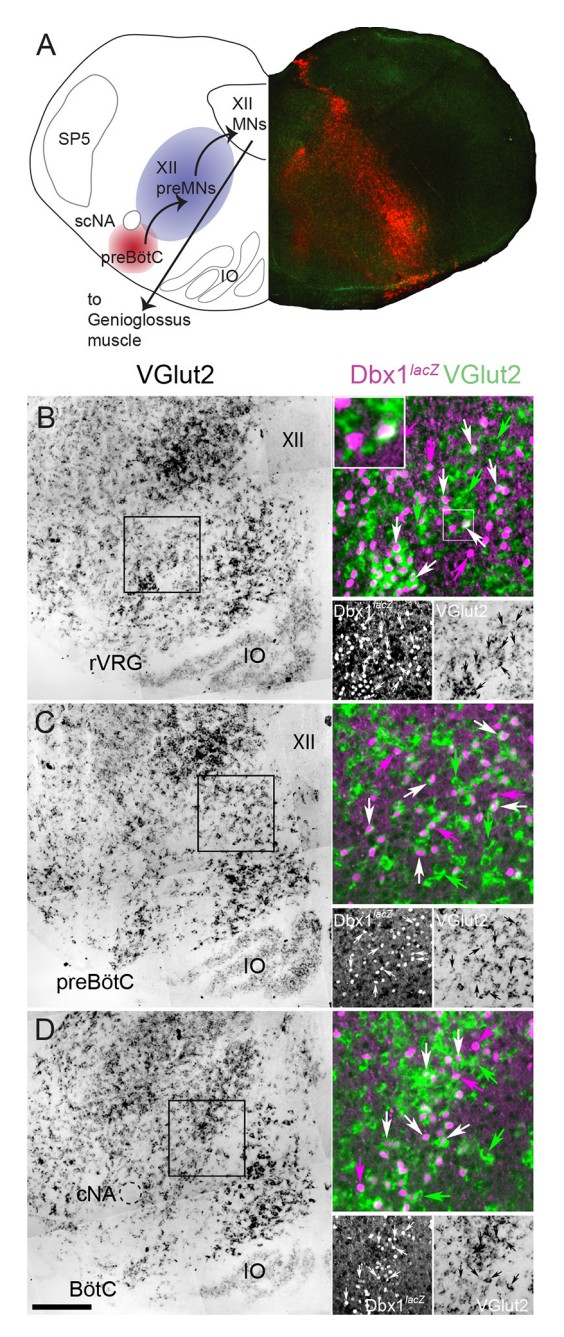

**Figure 1.** Fifty-seven percent of Dbx1 IRt neurons are glutamatergic. (**A**) Composite diagram of a neonatal mouse medullary slice showing (left) a schematic of the multi-synaptic medullary circuit where inspiratory rhythm in vitro is generated within the preBötC (red), transmitted to inspiratory premotoneurons (preMNs) in the IRt (purple), and hypoglossal motoneurons (XII MNs) in the XII nucleus that innervate the genioglossus muscle of the tongue. Right, fluorescent image of a transverse confocal section showing localization of Dbx1-derived cells (red) extending from the preBötC through the IRt. (**B-D**) Mosaic images of bright field in situ hybridization for VGlut2 in the IRt and ventral medulla at the level of the rVRG (corresponding to the -0.55 mm plate of Fig S1 [*Ruangkittisakul et al., 2014*]) (**B**), the preBötC (corresponding to the -0.45 mm plate of Fig S1 [*Ruangkittisakul et al., 2014*]) (**C**), and the BötC (corresponding to the -0.30 mm plate of Fig S1 [*Ruangkittisakul et al., 2014*]) (**D**) from a P0 *Dbx1^{lacz/+}* brainstem. Insets expanded in the right sides of panels **B-D** show confocal expression of *lacZ* gene product β-galactosidase (pseudocolor magenta, lower left) and VGlut2 (pseudocolor green, lower right) within the reticular formation with examples of glutamatergic Dbx1 reticular neurons (white arrows). The inset expanded in the upper left of panel B highlights one cell that shows colocalized *lacZ* and VGlut2 expression. Green arrows indicate

*Figure 1 continued on next page*

*Figure 1 continued*

glutamatergic neurons lacking *lacZ* co-expression. Magenta arrows indicate *lacZ*-expressing cells lacking VGlut2 co-expression. Scale bar = 250 µm. SP5 – trigeminal nucleus, cNA – compact division of nucleus ambiguus, scNA – semi-compact division of nucleus ambiguus, preBötC - pre-Bötzinger Complex, BötC – Bötzinger Complex, IO – inferior olive, rVRG – rostral ventral respiratory group, XII – hypoglossal nucleus.

(*Greer et al., 1991*; *Funk et al., 1993*; *Ireland et al., 2008*; *Kubin and Volgin, 2008*), we tested the hypothesis in vitro that Dbx1-derived IRt neurons are XII inspiratory premotoneurons that provide an important source of inspiratory drive to XII motoneurons.

## Results

Dbx1-expressing precursor cells are the primary source of glutamatergic neurons in the IRt (*Gray, 2013*); however, they also give rise to glycinergic, GABAergic and cholinergic neurons, and glia (*Gray et al., 2010*). To measure the percentage of glutamatergic Dbx1 IRt neurons we used in situ hybridization to identify vesicular glutamate transporter 2 (VGlut2, coded by the *Slc17a6* gene) in $Dbx1^{lacZ}$ reporter mice. Of the total 2777 *lacZ*-labeled cells we counted along the length of the medullary IRt (from the caudal end of the facial nucleus to the caudal pole of the lateral reticular nucleus (*Gray, 2008*, *2013*; *Ruangkittisakul et al., 2014*)) in four animals (694 ± 67 cells/animal), 1591 cells (398 ± 38 cells/animal) co-localized *Slc17a6*; i.e., 57 ± 1% of Dbx1 IRt neurons (*Figure 1B-D*) are glutamatergic.

We recorded 34 Dbx1 IRt neurons in rhythmically active slices obtained from $Dbx1^{CreERT2}$; $Rosa26^{tdTomato}$ mice that received synaptic drive potentials and generated volleys of actions potentials in phase with inspiratory-related XII nerve bursts (29 whole cell, 5 loose on-cell patch configuration) (*Figure 2A-C*). Six of the total 34 inspiratory modulated Dbx1 IRt neurons were recorded in a subset of experiments designed to estimate the proportion of Dbx1 IRt neurons that receive inspiratory drive. Counting all inspiratory and noninspiratory Dbx1 IRt neurons in this subset, we ascertained that 6 of 26 Dbx1 IRt neurons (23%) received inspiratory drive.

We investigated whether Dbx1 IRt neurons project to the XII nucleus. Ten of 23 (43%) inspiratory and five of 14 (36%) noninspiratory Dbx1 IRt neurons were antidromically activated via electrical stimulation of the ipsilateral XII nucleus. Two of the 10 antidromically activated inspiratory neurons also showed positive collision tests (*Figure 2D*). Second, anatomical reconstruction of seven biocytin-filled Dbx1 IRt inspiratory neurons revealed two neurons with axons that projected ipsilaterally towards the XII nucleus border, one of which projected into the XII nucleus and was antidromically activated (*Figures 2E, F*). Interestingly, two of these seven reconstructed inspiratory neurons were antidromically activated from the ipsilateral XII nucleus and had commissural axons, indicating bilateral connectivity. The remaining three reconstructed inspiratory Dbx1 IRt neurons had a commissural axon without antidromic activation (*Figure 2G*). *Figure 2H* shows the location of 24 of 34 Dbx1 inspiratory neurons we recorded. Criteria for inclusion in our putative premotoneuron sample were somatic location in the IRt, tdTomato expression, and inspiratory modulation. Additional evidence obtained in some cells included antidromic activation from the XII nucleus, positive collision tests, and axonal projections to the ipsilateral XII nucleus.

IRt neurons and rhythmogenic preBötC neurons (*Picardo et al., 2013*) are both derived from Dbx1 precursors (*Bouvier et al., 2010*; *Gray et al., 2010*). Their passive membrane properties are indistinguishable but the peak amplitude and area of inspiratory drive potentials are smaller in Dbx1 IRt compared to Dbx1 preBötC neurons (*Figure 3A-G*). In addition, the inspiratory drive potential in Dbx1 IRt neurons begins 90 ± 32 ms (n=13) prior to the onset of XII motor output (inset, green arrows, *Figure 3A,H*), which is later than drive potential onset in Dbx1 preBötC neurons (308 ± 17 ms prior to XII motor output (p=9E-7, unpaired t-test) (*Picardo et al., 2013*) (*Figure 3H*). Despite statistical differences, overlap between populations is such that no single parameter can definitively distinguish a Dbx1 preBötC neuron from a Dbx1 IRt neuron.

To directly test whether Dbx1 IRt neurons contribute inspiratory premotor drive to XII motoneurons, we sequentially laser-ablated Dbx1 neurons on one side, while recording XII nerve inspiratory output bilaterally. Ipsilateral to the ablation domain, XII burst magnitude decreased monotonically

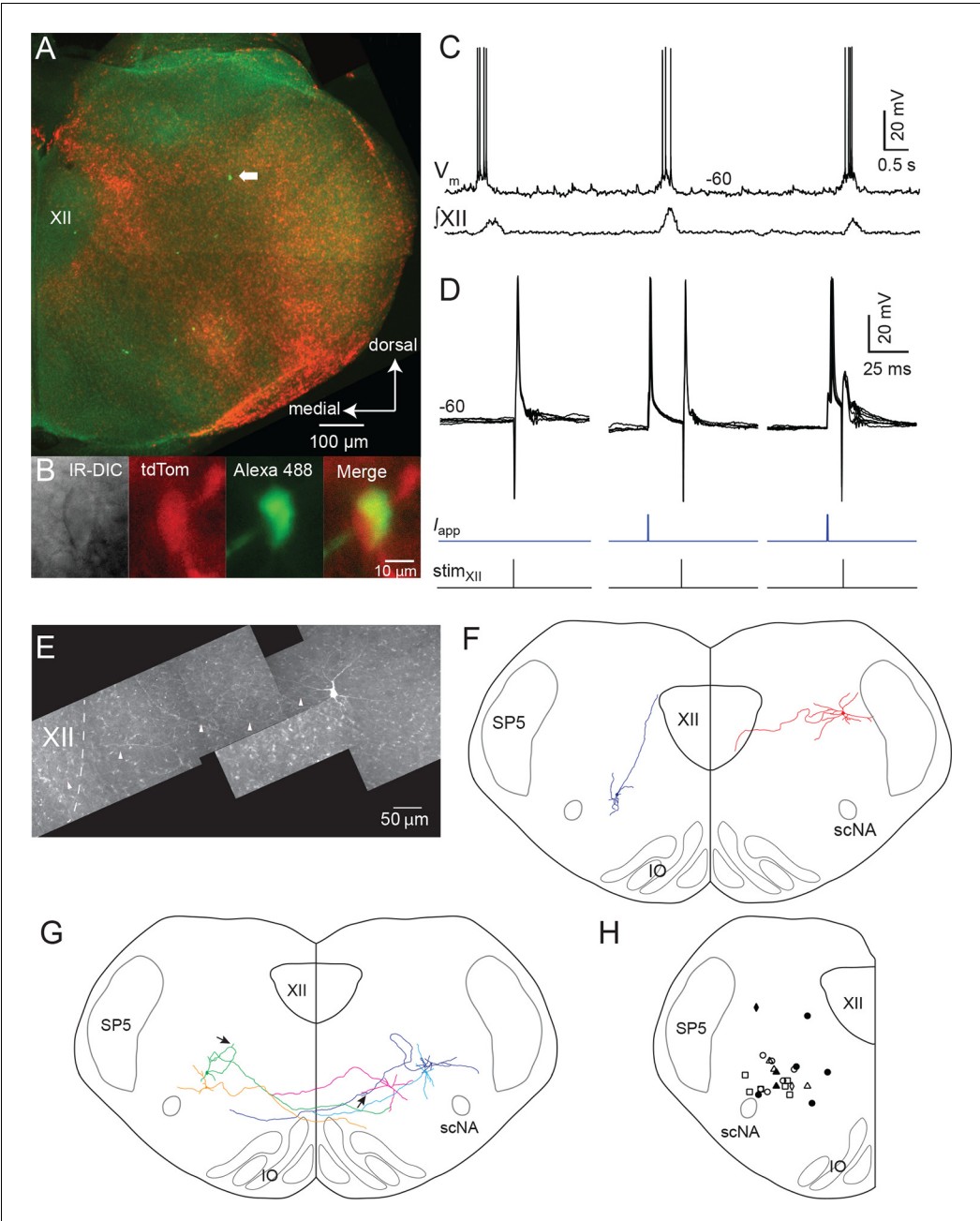

**Figure 2.** Inspiratory Dbx1 IRt neurons are ipsilaterally projecting, putative XII premotoneurons. (**A**) Right half of the rhythmic slice preparation after PFA fixation and optical clearing using the Scale method. White arrow points to the soma of the inspiratory-modulated Dbx1 neuron, which is further characterized in panels **B-E**. (**B**) From left to right, IR-DIC image of the recorded neuron and epifluorescence images showing tdTomato, Alexa 488 dialyzed from the intracellular solution, and a composite of tdTomato and Alexa 488 labeling. (**C**) Whole cell recording of imaged neuron showing rhythmic inspiratory firing (top trace) and integrated inspiratory XII nerve output (bottom trace). (**D**) Membrane potential (upper black traces), whole cell stimulation (0.8 mA, 1 ms duration, middle blue traces), and XII stimulation time (0.3 mA, 0.2 ms duration, bottom black trace, stimulation time is at vertical line); Left panel: antidromic action potentials activated from XII nucleus stimulation, 5 traces overlaid; Middle: evoked orthodromic action potentials followed 46 ms later by XII nucleus stimulation, 4 traces overlaid; Right: evoked orthodromic action potential followed 16 ms later by XII stimulation resulted in the extinction of the antidromic action potential (i.e., a successful collision test, 7 traces overlaid). (**E**) Composite diagram showing biocytin-filled, FITC-labeled, reconstructed neuron; arrowheads point to the axon. Orientation is the same as the image in A (images adjusted for brightness and contrast). (**F**) Morphologic reconstruction of all neurons that only projected ipsilaterally, illustrating axon trajectory and dendritic tree organization (n = 2). (**G**) Morphologic reconstruction of all commissurally-projecting neurons illustrating axon trajectory and dendritic tree organization (n = 5). Reconstructed neurons on the left-hand side were also antidromically activated. Arrowhead indicates axon bifurcation. (**H**) Image showing location for 24 of 34 recorded inspiratory Dbx1 IRt neurons. Open squares: antidromic activation not tested; open circles: negative antidromic

*Figure 2 continued on next page*

*Figure 2 continued*
activation; filled circles: positive antidromic activation; open triangles: commissural projection, negative antidromic activation; filled triangles: commissural projection, positive antidromic activation; open diamonds: ipsilateral projection, negative antidromic activation; filled diamond: ipsilateral projection, positive antidromic activation. XII – hypoglossal nucleus, IO – inferior olive, SP5 – trigeminal nucleus, scNA – semicompact division of nucleus ambiguus.

as the ablation tally increased, and remained stable once the ablation target list was exhausted (*Figure 4A*). Overall, ipsilateral, cell-specific laser ablation of Dbx1 IRt neurons in the superficial 100 μm of 500 μm slices significantly decreased ipsilateral XII burst amplitude and area by 36 ± 4% and 54 ± 3%, respectively. Cumulative ablation of Dbx1 IRt neurons did not perturb contralateral XII nerve output (*Figure 4C*) or inspiratory cycle period (*Figure 4D*).

## Discussion

For breathing, rhythmogenic interneurons and respiratory motoneurons are relatively well understood (*Rekling et al., 2000*; *Feldman and Del Negro, 2006*; *Feldman et al., 2013*; *Funk and Greer, 2013*), but physiological characterization of embryonically defined, functionally identified premotoneurons is lacking. Therefore, this study advances our previous work (*Hayes et al., 2012*; *Wang et al., 2013*; *Wang et al., 2014*) by demonstrating that Dbx1 IRt neuron ablation disrupts inspiratory burst amplitude but not rhythm. These data strongly suggest that a subpopulation of Dbx1-derived IRt neurons serve an important inspiratory premotor function.

While we did not perform dual intracellular recordings to unequivocally show synaptic contact between Dbx1 IRt neurons and XII motoneurons, our conclusion that Dbx1 IRt neurons are XII inspiratory premotoneurons is based on: (i) synaptic drive potentials and action potential volleys in Dbx1 IRt neurons that were in phase with inspiratory rhythm; (ii) evidence from antidromic activation (including positive collision tests) and axon reconstructions showing that inspiratory Dbx1 IRt neurons project to the XII nucleus; and, (iii) the sequential appearance, on average, of inspiratory-related activity first in preBötC neurons, then Dbx1 IRt neurons and finally in the XII nerve.

The significant reduction in ipsilateral inspiratory XII motor output following unilateral ablation of Dbx1 IRt neurons without a corresponding change in frequency also strongly supports an inspiratory premotor function. Nonspecific ablation-associated damage to circuits contained in the slice is unlikely to be a factor since tissue-matched control slices similarly subjected to cumulative laser ablation protocols showed no degradation in previous reports (*Hayes et al., 2012*; *Wang et al., 2013*; *Wang et al., 2014*).

However, there are two main limitations to the ablations experiments. First, Dbx1 IRt neurons detected for ablation included inspiratory and noninspiratory neurons. Selective ablation of inspiratory-modulated, Dbx1 IRt neurons with demonstrated synaptic connections to XII motoneurons would be required to more thoroughly assess the significance of Dbx1 IRt inspiratory XII premotoneurons. We predict a more precipitous, and possibly complete, decline in inspiratory burst amplitude if bona fide Dbx1 IRt inspiratory premotoneurons could be selectively ablated, avoiding Dbx1 IRt neurons with alternative functions.

A second limitation is that the ablated Dbx1 IRt neurons have unknown transmitter phenotype. Just over half of Dbx1 IRt neurons are glutamatergic. Thus, the ablated population will include GABA and glycinergic neurons. This limitation, however, is unlikely to impact significantly our conclusion that Dbx1 IRt neurons are a major source of excitatory (glutamatergic) inspiratory XII premotor drive. Phrenic motoneurons in vitro (*Parkis et al., 1999*) and XII motoneurons in vivo (*Woch and Kubin, 1995*) receive weak concurrent inspiratory inhibitory premotor drive, but there is no evidence of tonic (*Numata et al., 2012*), nor endogenous inspiratory inhibitory XII premotor drive in the slice. Rhythmic inspiratory XII motoneuron drive in the rhythmic slice preparation, and therefore premotor activity, is entirely glutamatergic (*Greer et al., 1991*; *Funk et al., 1993*; *Ireland et al., 2008*; *Kubin and Volgin, 2008*). Moreover, while XII motoneurons and XII inspiratory output in vitro are inhibited by exogenous activation of GABA and glycine receptors (*O'Brien and Berger, 1999*; *Jaiswal et al., 2015*), ablation of endogenously active inhibitory premotoneurons, tonic or phasic, would potentiate rather than reduce XII inspiratory output. Thus, in the unlikely circumstance that

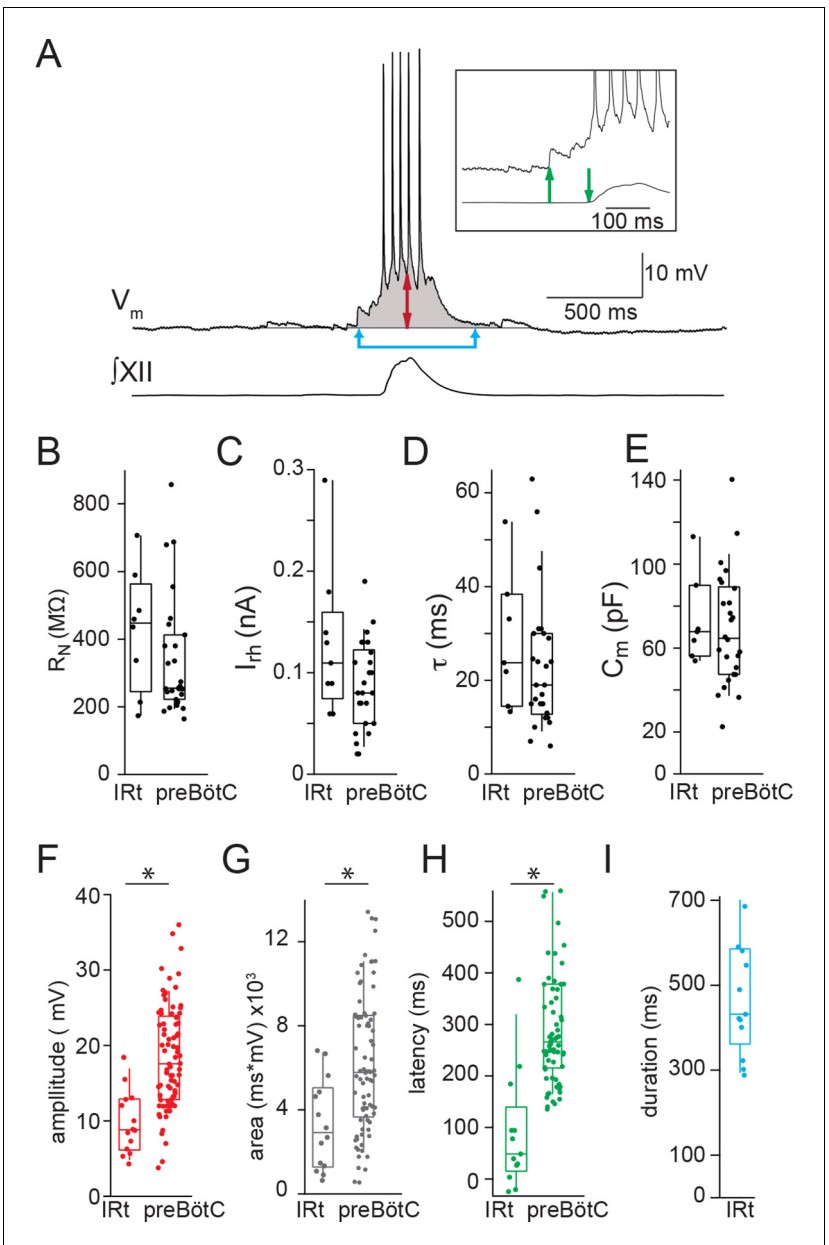

**Figure 3.** Electrophysiological characteristics of Dbx1 IRt neurons. (**A**) Membrane potential recording ($V_m$) from an inspiratory Dbx1 IRt neuron and integrated XII nerve (∫XII) activity showing inspiratory burst characteristics. Inspiratory drive amplitude for neurons that generated action potentials during inspiratory bursts was estimated based on the shape of the underlying drive potential (double-ended red arrow). Inspiratory drive area was calculated as the integral of membrane potential over time (shaded area). Panel inset: inspiratory drive latency was defined as the delay between the onset of inspiratory depolarization (upward green arrow) and the onset of XII inspiratory nerve burst (downward green arrow). Inspiratory drive duration was measured as the length of time the membrane potential was above baseline (joined blue arrows). Membrane potential scale bar applies to inset as well. Group data (median, box: interquartile range, whiskers: 10th and 90th percentiles) and individual values (solid circles) measuring passive membrane properties and inspiratory drive characteristics in Dbx1 neurons of the IRt and preBötC: (**B**) neuronal input resistance, $R_N$, n = 8 (IRt), n = 27 (preBötC); (**C**) rheobase, $I_{rh}$, n = 9 (IRt), n = 26 (preBötC); (**D**) membrane time constant, $\tau$, n = 7 (IRt), n = 26 (preBötC); (**E**) whole-cell capacitance, $C_m$, n = 7 (IRt), n = 26 (preBötC); (**F**) inspiratory drive amplitude, n = 14 (IRt), n = 82 (preBötC); (**G**) inspiratory drive area, n = 14 (IRt), n = 82 (preBötC); (**H**) inspiratory drive latency, n = 13 (IRt), n = 70 (preBötC); (**I**) inspiratory drive duration, n = 13 (IRt); All preBötC data from (*Picardo et al., 2013*). IRt – intermediate reticular formation. *, $p < 0.05$, unpaired t-test.

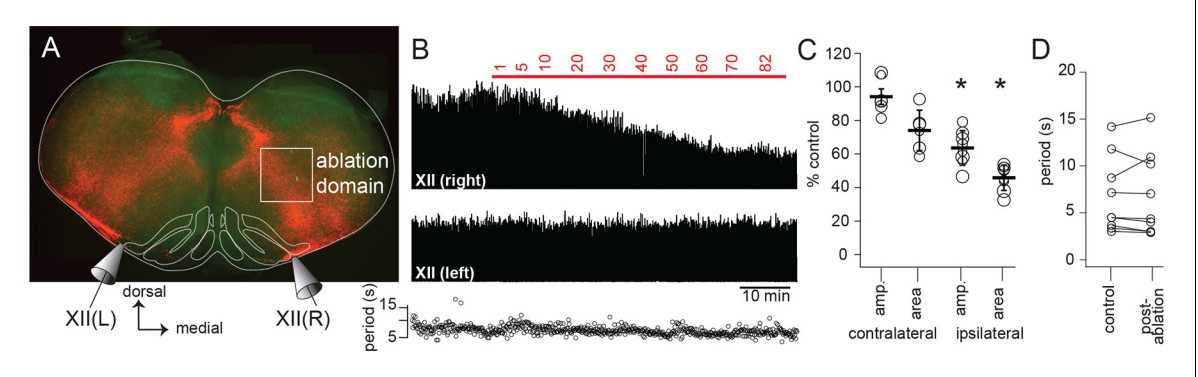

**Figure 4.** Dbx1 IRt neurons contribute significantly to XII inspiratory burst output. (**A**) Confocal image of a rhythmic slice preparation after fixation and Scale clearing illustrating the region targeted for laser ablations within the white rectangle on the right-hand side. Bilateral XII output was recorded, XII (R) and XII(L). (**B**) Example bilateral XII output, corresponding to the slice in (**A**). Integrated XII output corresponds to the lesioned side [XII(right)] and the non-lesioned side [XII(left)] as a function of time and lesion number. The red bar indicates the start and duration of ablations and the numbers above show the confirmed ablation tally. (**C**) Group data (mean ± SEM) and individual values (open circles) for contralateral (i.e. unlesioned) XII amplitude and area, n = 6; and for ipsilateral (i.e. lesioned) XII amplitude and area, n = 8. *, p < 0.05, comparison of XII amplitude or area pre- and post-ablation, repeated measures ANOVA, Tukey post-hoc test, n = 5. (**D**) Individual values (open circles) pre- and post-ablation, linked by solid lines, of XII inspiratory burst period. IRt – intermediate reticular formation.

we ablated endogenously active inhibitory Dbx1 IRt premotoneurons, it would underestimate rather than contradict our conclusion that Dbx1 IRt premotoneurons provide excitatory drive for inspiratory motor output.

We identified at least three types of Dbx1 neurons in the IRt: (i) inspiratory premotoneurons that project ipsilaterally; (ii) inspiratory premotoneurons that project ipsi- and contralaterally; and, (iii) noninspiratory premotoneurons. Inspiratory neurons comprise more than 20% of Dbx1 IRt neurons. Ipsilaterally projecting Dbx1 IRt premotoneurons are a significant source of glutamatergic inspiratory drive to XII motoneurons in vitro. Whether they are the sole source is uncertain. Persistence of inspiratory XII nerve activity following laser ablation of Dbx1 IRt neurons might suggest additional non-Dbx1 derived sources of inspiratory premotor drive. However, the ablation-resistant activity could also derive from Dbx1 premotoneurons too deep for laser ablation, or Dbx1-derived neurons that went untargeted due to lack of tdTomato expression (see Methods). The contribution of ipsilaterally projecting Dbx1 IRt premotoneurons to inspiratory drive in vivo is an important question that cannot be extrapolated from in vitro experiments. Portions of the XII premotor network, as well as sources of modulatory drive, are missing from slices, which may make inspiratory XII output more sensitive to the loss of premotoneurons.

Contralateral projections from Dbx1 IRt inspiratory putative premotoneurons were unexpected and of unknown function. They may contribute to the contralateral inspiratory drive for airway and tongue movements (*Ono et al., 1994*; *Peever et al., 2002*), they may be inspiratory rhythmogenic preBötC neurons displaced into the IRt, or they may help coordinate inspiration with other oromotor behaviors (*Ono et al., 1998*; *Tupal et al., 2014a*).

From a clinical perspective, XII inspiratory premotoneurons are relevant in the context of obstructive sleep apnea where reductions in airway muscle tone, especially in the genioglossus muscle of the tongue, during sleep can result in airway obstruction and apnea. XII inspiratory premotoneurons decrease their firing during carbachol-induced REM sleep (*Woch et al., 2000*). Thus, understanding the modulatory control of XII premotoneuron excitability during sleep-wake cycling may provide insight into mechanisms underlying sleep-disordered breathing.

## Methods

### Ethics approval

All the experiments were performed in accordance with guidelines laid down by the NIH in the US regarding the care and use of animals for experimental procedures, the Institute for Laboratory Animal Research (*National Academy of Sciences, 2010*; *National Academy of Sciences, 2015*), and in compliance with protocols approved by the College of William and Mary Institutional Animal Care and Use Committee, the Animal Studies Committee at Washington University School of Medicine and the University of Alberta Faculty of Medicine Animal Welfare Committee.

### Animals

We used Cre-Lox transgenic technology to identify Dbx1 interneurons via native fluorescent protein expression. We crossed female mice that express Cre recombinase fused to the tamoxifen-sensitive estrogen receptor ($Cre^{ERT2}$) under the control of the *Dbx1* promoter, i.e., $Dbx1^{CreERT2}$ with floxed male reporters that express the red fluorescent protein tdTomato in a Cre-dependent manner ([$B6;129S6\text{-}Gt(ROSA)26Sor^{tm9(CAG-tdTomato)Hze}$/], Jax no. 007905) ($Rosa26^{tdTomato}$) (*Madisen et al., 2010*). Offspring with both alleles ($Dbx1^{CreERT2}$;$Rosa26^{tdTomato}$ mice) express the fluorescent reporter in Dbx1-derived cells (*Hirata et al., 2009*; *Picardo et al., 2013*; *Ruangkittisakul et al., 2014*). The time of conception, embryonic day 0.5 (E0.5), was defined as 24 hr following the start of cohabitation. Pregnancy was confirmed by subsequent weight gain. For the in situ hybridization experiments, we utilized $Dbx1^{lacZ/+}$ mice crossed and bred on a C57BL6 or mixed CD1/C57BL6 background (*Pierani et al., 2001*). Mice were genotyped by PCR using primers specific to $Dbx1^{lacZ}$ as previously described (*Pierani et al., 2001*). A total of 43 mice were used for experiments.

### Preparations

Dbx1 progenitor cells are born from approximately E9.5 to E12.5 (*Pierani et al., 2001*; *Hirata et al., 2009*; *Gray et al., 2010*). To generate experimental animals with fluorescently labeled Dbx1 neurons, we administered Tamoxifen (T5648; Sigma Aldrich, St Louis, MO, dissolved in corn oil at 5 mg/ml) via oral gavage (1 mg/40 g body weight) at E10.5. Thus, even if the estimated conception date was off by 24 hr, tamoxifen was still administered during the early stages of Dbx1 endogenous expression, (*Pierani et al., 2001*; *Hirata et al., 2009*; *Gray et al., 2010*). The timing of tamoxifen administration influences the proportion of Dbx1-derived neurons that express tdTomato, which will contribute to variable tdTomato labeling (e.g., *Figures 1A*, *2A* and *4A*). Note also that Tamoxifen-mediated Cre recombination is not completely efficient; some Dbx1-derived neurons will not express tdTomato.

$Dbx1^{CreERT2}$;$Rosa26^{tdTomato}$ neonatal mouse pups (post-natal day 0-5, P0-5) of either sex were used to generate slice preparations that retain minimal respiratory rhythm- and pattern-generating circuits (*Smith et al., 1991*; *Funk and Greer, 2013*; *Ruangkittisakul et al., 2014*). Neonatal mice were anaesthetized via immersion hypothermia, which achieved a similar level of insentience compared to inhaled or injectable anesthetic agents (*National Academy of Sciences, 2013*). Mice were immersed in crushed ice until the withdrawal reflex disappeared. The skin overlying the skull was then removed, the animal decerebrated, and then transected at the caudal margin of the ribcage. The remaining skull and thorax were immersed in ice-cold, oxygenated (95% $O_2$, 5% $CO_2$) artificial cerebral spinal fluid (aCSF). The brainstem-spinal cord was then isolated, glued to an agar block and placed in the vise of a vibratome (Thermo Scientific Microm HM 650 V, Waltham, MA) with the ventral surface facing the cutting edge. The preparation was oriented at a ~10 degree less steep than used to prepare the calibrated rhythmic slice preparation (*Ruangkittisakul et al., 2014*), which exposed Dbx1 IRt neurons with inspiratory activity on the slice surface. Serial transverse sections were cut in the rostral to caudal direction until the most rostral portion of the inferior olive was visible at the rostral face of the neuraxis (i.e., approximately 0.17 mm from the caudal facial nucleus, -0.15 mm slice, Fig. S1, *Ruangkittisakul et al., 2014*). A single 500-μm-thick section containing the preBötC, XII inspiratory premotoneurons, XII motor nuclei and rostral XII nerve rootlets was isolated and placed caudal surface up in the recording chamber of a fixed-stage upright microscope (Zeiss Axioskop 2 FS plus, Thornwood, NY, USA) equipped with bright field, and infrared differential interference contrast (IR-DIC) and epifluorescence microscopy. The preparation was held in place using a

platinum frame with nylon fibers. The chamber was continuously perfused (4 ml/min) with warmed (27-28°C) oxygenated artificial cerebral spinal fluid (aCSF).

## Solutions

Standard aCSF used for dissection contained (in mM): 124 NaCl, 3 KCl, 1.5 CaCl$_2$, 1 MgSO$_4$, 25 NaHCO$_3$, 0.5 NaH$_2$PO$_4$ and 30 dextrose, equilibrated with 95% O$_2$ and 5% CO$_2$, which resulted in pH of 7.4. The aCSF [K$^+$] was raised to 9 mM for electrophysiological recordings. Although elevated K$^+$ is unnecessary for inspiratory rhythm generation in slices (*Ruangkittisakul et al., 2006*), it prolongs robust rhythmic function to facilitate the prolonged protocols.

Intracellular solution used for whole-cell patch-clamp recordings contained (in mM): 140 potassium gluconate, 10 HEPES, 5 NaCl, 1 MgCl$_2$, 0.1 EGTA, 2 Mg-ATP, 0.3 Na-GTP. Alexa 488 hydrazide (A10437; Invitrogen, Carlsbad, CA, 50 µM) was added to the intracellular solution for fluorescent visualization of neurons during the recording. Biocytin (B4261; Sigma Aldrich, St. Louis, MO, 2 mg ml$^{-1}$) was added to the intracellular solution to facilitate mapping of neuron location and tracing of axonal projection patterns after the experiment.

## Electrophysiology

Inspiratory burst output was recorded from the XII nerve rootlets using suction electrodes connected to a differential amplifier (Dagan Instruments, Minneapolis, MN, 2000x, 300-1000 Hz band-pass filter). This signal was full-wave rectified and smoothed (50 ms smoothing window) for analysis of burst frequency and amplitude. Pipettes used for whole-cell recording were fabricated from borosilicate glass (OD: 1.5 mm, ID: 0.87 mm, Harvard Apparatus, Edenbridge, UK) to a tip resistance of 4-6 MΩ. Whole-cell current-clamp recordings were obtained under visual guidance with optics for IR-DIC microscopy using a Dagan IX2-700 current-clamp amplifier (Dagan Instruments). Recordings were low-pass filtered at 1 kHz and digitally acquired at 4-10 KHz using a PowerLab 16-bit A/D converter (AD Instruments, Colorado Springs, CO). All protocols in Dbx1 neurons were initiated from a holding potential of -60 mV.

Neurons were targeted for recording based on native tdTomato expression and anatomical location within the IRt, which is bounded by a region dorsomedial to nucleus ambiguus and ventrolateral to the XII nucleus.

An antidromic activation protocol was used to test whether Dbx1 IRt neurons project to the XII nucleus. A bipolar concentric stimulating electrode (catalogue #: CBBRF50, FHC, Bowdoin, ME) was placed on the surface of the XII nucleus. A whole-cell current-clamp recording was then obtained from an ipsilateral, inspiratory Dbx1 IRt neuron and cathodic stimulation (0.2 ms duration, 1 Hz frequency) was applied with increasing intensity (0.05-0.4 mA, controlled by a stimulus isolation unit) until either a short latency antidromic action potential was observed, or the maximum current (0.4 mA) was reached. If an antidromic action potential was evoked, then a collision test was performed in which an orthodromic action potential was first activated via 1-ms step current commands applied to the whole-cell pipette. Antidromic stimulation was then applied with progressively decreasing delays until the antidromic action potential recorded at the soma of the Dbx1 IRt neuron disappeared, indicating a successful collision test.

## Reconstruction of neuron location and axonal trajectory

To map neuron location and characterize axonal projection patterns, whole-cell recordings of inspiratory Dbx1 IRt neurons were maintained for a minimum of 20 min to facilitate diffusion of biocytin into the neuron. When the experiment was over, the slice was fixed in a 4% (w/v) paraformaldehyde solution in 0.1 M Na-Phosphate buffer at 4°C for at least 16 hr. Slices were then incubated for 10 days in Scale solution (*Hama et al., 2011*), which contained 4 M urea, 10% (w/v) glycerol and 0.1% (w/v) Triton X-100, in order to remove fat and increase tissue transparency. Following the Scale procedure, slices were rinsed in phosphate-buffered saline (PBS) (1 hr), PBS and 10% heat-inactivated fetal bovine sera (FBS, F4135; Sigma-Aldrich) (15 min), followed by PBS + FBS and 1% Triton X-100 (45 min). Slices were then incubated on a nutator overnight at 4°C in fluorescein-isothiocyanate (FITC)-conjugated ExtrAvidin (E2761; Sigma-Aldrich), and then washed six times in PBS for 20 min each. Slices were then placed on slides, cover-slipped in Vectashield (H-1400 Hard Set, Vector Laboratories, Burlingame, CA), and Dbx1 IRt neurons were visualized using a spinning-disk confocal

microscope (Olympus BX51, Center Valley, PA) or a laser-scanning confocal microscope (Zeiss LSM 510, Thornwood, NY). A low-magnification image of the entire slice was generated using a 4X objective (numerical aperture [NA]: 0.13) and a 20X objective (NA: 0.5) was used to obtain higher resolution z-stack images of the labeled neurons.

Neuron location was determined based on the analysis of the 4X images, which showed the position of the labeled neurons relative to relevant landmarks, which included the inferior olive, nucleus ambiguus, XII nerve roots, and the spinal trigeminal nucleus. Dbx1 neurons located dorsal to the dorsal border of nucleus ambiguus were defined as part of the IRt, and therefore putative premotoneurons. The dorsal border of nucleus ambiguus was defined by visual inspection of the slice. When this was not visible, the dorsal border of nucleus ambiguus was defined based on relative position along the dorsoventral (vertical) axis of the slice. Using the brain atlas developed for the $Dbx1^{CreERT2};Rosa26^{tdTomato}$ mouse strain used here (*Ruangkittisakul et al., 2014*), we drew a line over the slice lateral to midline to identify the maximum length of the slice along the dorsoventral axis. A second line was drawn horizontally from the dorsal border of the nucleus ambiguus, which generally coincided with the dorsal limit of the inferior olive, so that it intersected perpendicular to the dorsoventral transect. Based on this analysis of the published atlas and our own sections, the dorsal border of nucleus ambiguus was located 36-37% of the maximum length along the ventrodorsal axis (closer to the ventral surface). Position along the mediolateral axis was based on visual inspection of the labeled neuron relative to local landmarks as well as relative position along the mediolateral axis. The locations of all neurons were mapped onto summary schematic diagrams. An inspiratory neuron was considered part of the IRt if it was in the Dbx1 cell column and in the dorsal 63% of the slice; and had a somal diameter >10 µm. Some glial cells derive from a Dbx1 lineage, but these are typically <10 µm (*Gray et al., 2010*). Only Dbx1 neurons receiving rhythmic inspiratory synaptic inputs were analyzed.

*Axonal Trajectory.* The z-stack images taken at 20X magnification and 1-µm increments were iteratively stitched together using the 3D Stitching Plugin for ImageJ (*Preibisch et al., 2009*). Brightness and contrast were adjusted. Adobe Photoshop (Adobe Systems, San Jose, CA) was used to create the composite image of the entire slice at 4X magnification, by pseudocolouring individual images and then manually aligning them. Finally, we used the open-source software Neuromantic (*Myatt et al., 2012*) to digitally reconstruct neuronal morphology, focusing on axon trajectory for the present analysis (*Parekh and Ascoli, 2013*).

## Tissue processing

*Dbx1^{lacZ/+}* neonatal pups (P0) were anesthetized and perfused (transcardiac) with 4% paraformaldehyde (PFA) in 0.1 M PBS at pH 7.4, postfixed in PFA overnight at 4°C, cryoprotected in 25% sucrose in PBS, blocked, frozen in OCT, and stored at -75°C. Hindbrains were sectioned in sets of six on a Hacker (Winnsboro, SC) cryostat at 20 µm and sections were thaw-mounted onto Superfrost Plus (Fisher Scientific, Hampton, NH) slides and stored at -20°C until ready for in situ hybridization and immunohistochemical protocols.

## In situ hybridization

Slides were removed from -20°C storage, immersed in 4% PFA in 0.1 M PBS, permeabilized with radio-immunoprecipitation assay buffer (RIPA) buffer, washed in 0.1 M triethanolamine-HCl with 0.25% acetic anhydride, blocked in hybridization buffer at 65°C, and then placed into slide mailers containing hybridization buffer with digoxigenin labeled antisense vesicular glutamate transporter 2 (VGlut2) cRNA at 1 µg/ml overnight at 65°C (*Tupal et al., 2014b*). Slides were washed in sodium citrate buffers at 62°C, then washed and incubated in alkaline phosphatase conjugated anti-DIG antibody in 10% normal horse serum and incubated in nitro blue tetrazolium chloride and 5-Bromo-4-chloro-3-indolyl phosphate (NBT-BCIP, Roche, Indianapolis, IN) until cellular labeling was clear. Slides were processed for mRNA expression prior to immunohistochemical labeling as previously described (*Gray, 2013*). All compounds were acquired from Sigma-Aldrich (St. Louis, MO).

## Immunohistochemistry

Slidemounted tissue sections were then washed in PBS with 0.2% triton X-100, blocked in 10% heat-inactivated normal horse sera, incubated in chicken anti-beta galactosidase antibody (*lacZ*) 1:1000

(Abcam, Cambridge, MA) overnight at 4°C, incubated in secondary antibody for 2 hr at room temperature, or at 4°C overnight, and coverslipped in Vectashield (Vector Laboratories, Burlingame, CA).

## Immunohistochemistry and in situ hybridization image acquisition

Fluorescent and brightfield images were acquired using a Nikon 90i microscope (Nikon Instruments, Melville, NY), Roper H2 -cooled CCD camera (Photometrics, Tucson, AZ), and Optigrid -Structured Illumination Confocal with a Prior (Rockland, MA) motorized translation stage. Pseudocolored images were acquired in Volocity (PerkinElmer, Waltham, MA), filtered in Photoshop with the noise, dust and scratches filter (radius 2, threshold 3), and modified for clarity by adjusting levels to use the full greyscale range in Photoshop or ImageJ (NIH, Bethesda, MD) (*Schneider et al., 2012*) and exported as 8-bit JPEG images.

## Cell counts

*lacZ*-expressing nuclei larger than 5 μm in diameter were counted along the length of the IRt (extending from the caudal end of the facial nucleus to the caudal pole of the lateral reticular nucleus (*Gray, 2008*, *2013*; *Ruangkittisakul et al., 2014*)) by visual inspection of mosaic images of combined VGlut2 mRNA in situ hybridization and confocal β-Gal immunohistochemistry acquired at 10X from the region dorsal to the nucleus ambiguus and 350 μm lateral to the midline. Dbx1-labeled cells within 175 μm of the midline were excluded because there are few XII premotoneurons in this region (*Dobbins and Feldman, 1995*). Levels of digital images were adjusted so that the full 255 range of greyscale levels were used, which served to maximize low-level expression. Co-localization of β-Gal and mRNA required at least half of the nucleus to be surrounded.

## Laser ablation

Cell-specific detection and laser ablation of Dbx1 IRt neurons was carried out by an automated system previously applied to Dbx1 preBötC neurons, consisting of a Zeiss LSM 510 laser-scanning head and fixed-stage microscope body, an adjustable wavelength 1.5-W Ti-sapphire tunable laser, a robotically-controlled xy translation stage and custom software (*Hayes et al., 2012*; *Wang et al., 2013*; *Wang et al., 2014*). Detection and ablation procedures consist of three phases: (1) initialization, (2) target detection, and (3) ablation. The initialization phase defines the domain for detection and ablation, which in this case comprises a region of the IRt, as defined above, that is a 412 $\mu m^2$ in the transverse plane and 100 μm along the rostrocaudal axis; i.e., ablation was limited to neurons within 100 μm of the slice surface. This spanned the area dorsomedial to nucleus ambiguus and ventrolateral to XII nucleus, which excludes the preBötC. During the detection phase, tdTomato emission was detected using a HeNe 543 nm laser at depths <100 μm, while scripted routines making use of ImageJ functionality were used to differentiate Dbx1 neurons based on threshold-crossing algorithms (*Wang et al., 2013*). During the ablation phase, individual Dbx1 IRt neurons were randomly selected from the list of validated targets and scanned over a 10 $\mu m^2$ area at the center of the defined cell location by maximum-strength 800-nm Ti:Sapphire laser pulses. The time between ablations was ~30 s, which includes scanning time required for the ablation and for confirmation of the ablation. Lesions were confirmed by optical criteria described elsewhere and confirmed ablations were added to the running tally (*Wang et al., 2013*). Lesions were performed until the entire target list was exhausted. XII motor output was monitored for 30 min prior to beginning ablations and then throughout the ablations. XII motoneuron axons travel across the slice virtually parallel to the cut surface and then exit the brainstem through the ventral surface in multiple nerve rootlets that span more than 500 μm rostrocaudally. We record from the XII nerve root closest the targeted slice surface since it is most likely to contain axons of motoneurons that receive input from the surface Dbx1 neurons that we are able to ablate.

XII burst amplitude and area were computed using LabChart PeakParameters module and averaged over a 5-min window immediately prior to commencing target ablation (control) and a 5-min window after exhausting the target list (post-ablation). Inspiratory burst period was calculated from XII burst peak to peak throughout the entire experiment.

## Data analysis

All intrinsic cellular properties were measured in current clamp from a membrane potential of -60 mV, only including cells with spikes that reached at least 0 mV. Input resistance ($R_N$, M$\Omega$) was calculated from the slope of the voltage-current relationship from a series of 500-ms current pulses that hyperpolarize the membrane potential as much as 30 mV from the -60 mV baseline potential. The membrane time constant ($\tau$, ms) was computed from an exponential fit to membrane potential relaxation. Cell capacitance ($C_m$, pF) was calculated from the quotient of the time constant and the input resistance. Finally, rheobase ($I_{rh}$, pA) was measured by applying 3-ms depolarizing pulses that were manually adjusted until reaching the minimum pulse magnitude that elicited a single action potential on the termination of the current pulse. A minimum of five measurements was averaged for each parameter for each cell.

Measurements of the underlying inspiratory drive were computed in LabChart, and were averaged over at least six cycles. Inspiratory burst latency (drive latency) quantifies the time interval between summating excitatory post-synaptic potentials (EPSPs) in the recorded neuron (causing the neuron to depolarize and potentially fire action potentials) and the XII motor output; positive numbers indicate that the Dbx1 neuron depolarized before the onset of the XII root output. Drive latency was measured from the onset of summating EPSPs in the recorded neuron to the maximum slope of the XII burst (*Rekling et al., 1996*; *Picardo et al., 2013*). Drive amplitude was measured as the difference between maximum depolarization and the baseline (baseline defined as the average membrane potential during 250 ms of the interburst interval preceding the burst to be measured), if the depolarizing burst was insufficient to lead to action potentials. If, however, the neuron spiked during the inspiratory drive, the maximum depolarization was estimated based on the shape of the underlying drive potential. Drive potential area was measured as the integral of the burst, which (unlike drive potential amplitude) is a measure insensitive to intraburst spiking. Finally, drive potential duration was measured for both the XII root output and for inspiratory-modulated Dbx1 IRt neurons based on the elapsed time from when XII discharge or the membrane potential exceeded 10% of peak depolarization to the moment XII discharge or membrane potential dropped below 10% of peak depolarization.

## Acknowledgements

NIH grants R01-HL104127 (PI: Del Negro), R01-HL089742 (PI: Gray); CIHR grant 130306, RES0018140 (PI: Funk), Natural Sciences and Engineering Research Council (NSERC, 402532, PI: Funk), Alberta Innovates Health Solutions (AIHS), Canadian Foundation for Innovation (CFI), Alberta Science and Research Authority (ASRA), Women and Children's Health Research Institute, Stollery Children's Hospital Foundation.

## Additional information

### Funding

| Funder | Grant reference number | Author |
|---|---|---|
| National Heart, Lung, and Blood Institute | R01-HL089742 | Paul A Gray |
| Canadian Institutes of Health Research | Operating grant, 130306, RES0018140 | Gregory D Funk |
| Natural Sciences and Engineering Research Council of Canada | Discovery grant, 402532, RES0012299 | Gregory D Funk |
| Alberta Innovates - Health Solutions | Salary award | Gregory D Funk |
| Canada Foundation for Innovation | Equipment and infrastructure grant | Gregory D Funk |
| National Heart, Lung, and Blood Institute | R01-HL104127 | Christopher A Del Negro |

The funders had no role in study design, data collection and interpretation, or the decision to submit the work for publication.

## Author contributions
ALR, NCV, PAG, Conception and design, Acquisition of data, Analysis and interpretation of data, Drafting or revising the article; VTA, AK, Acquisition of data, Analysis and interpretation of data, Drafting or revising the article; CADN, GDF, Conception and design, Analysis and interpretation of data, Drafting or revising the article

## Author ORCIDs
Ann L Revill, http://orcid.org/0000-0002-6071-6866
Nikolas C Vann, http://orcid.org/0000-0003-4139-0642
Victoria T Akins, http://orcid.org/0000-0002-0061-7612
Andrew Kottick, http://orcid.org/0000-0002-3731-5140
Christopher A Del Negro, http://orcid.org/0000-0002-7848-8224
Gregory D Funk, http://orcid.org/0000-0001-5848-0631

## Ethics
Animal experimentation: Ethics Statement: All experiments were performed in accordance with guidelines laid down by the NIH in the US regarding the care and use of animals for experimental procedures, the Institute for Laboratory Animal Research, and in compliance with protocols approved by the College of William & Mary Institutional Animal Care and Use Committee (protocol #8828), the Animal Studies Committee at Washington University School of Medicine (protocol #20110249) and the University of Alberta of Medicine Animal Welfare Committee (protocol #255).

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
