## [Decision Letter]

Thank you for submitting your work entitled "Dbx1 precursor cells are a source of inspiratory XII premotoneurons" for consideration by *eLife*. Your article has been favorably evaluated by Eve Marder (Senior editor), a Reviewing editor (Ole Kiehn), and three reviewers.

The following individuals involved in review of your submission have agreed to reveal their identity: Jeff Smith and Nino Ramirez (peer reviewers).

The reviewers have discussed the reviews with one another and the Reviewing editor has drafted this decision to help you prepare a revised submission.

Summary:

In this Research Advance, Revill et al. present data showing that a subpopulation of Dbx1-derived neurons in the intermediate reticular formation (IRt) of the mouse medulla exhibit rhythmic inspiratory activity in neonatal slice preparations in vitro, and that some of these inspiratory neurons have axonal projections to the XII motor nucleus and are proposed to be inspiratory XII premotoneurons. The authors show that laser ablation of a set of ~70 Dbx1-tdTomato labeled neurons in the IRt significantly reduces the amplitude of the ipsilateral inspiratory hypoglossal motor output without affecting the frequency of the rhythm, thereby providing functional evidence that at least some of these DBx1 positive cells in this region of brainstem are premotor neurons downstream of the rhythm-generating inspiratory kernel mediating drive to the XII motor neurons. The authors argue that the novelty the present study is establishing the developmental origin (from Dbx1-expressing progenitors) of some identified inspiratory premotoneurons. Together with previous work showing that a subset of Dbx1-derived pre-BötC neurons are inspiratory neurons and function in rhythmogenesis, the authors also argue that the potential advance is the demonstration that that these premotor neurons are distinct from the inspiratory rhythm generating neuronal population but are derived from the same progenitors. The responses of the reviewers were generally positive and agree that study is a significant advance of the previous study of Dbx1 derived neurons in pre-BötC. The reviewers felt however that the authors' statement of the advance is out of line with what they actually show and that it contains overstatements for which there is no support in the presented data. While the study is technically sound and well performed, the authors need to acknowledge the limitations of their findings. These points are outlined below.

Essential revisions:

1) It is hard to see why is it a significant advance for understanding respiratory circuit function per se to know that a subset of IRt XII inspiratory premotor neurons are Dbx1-derived. Or that Dbx1-expressing progenitors give rise to both rhythm and pattern generating neurons. Knowing the genetic specificity of these cells does not give insight into how genetically defined neuronal populations determine function and connectivity. Rather the advance and novelty of the present study with respect to the previous study is the functional demonstration of a group of cells that are premotor (amplitude modulation not affecting frequency) different from the rhythm generation kernel (frequency and amplitude modulation). The authors should state that the main novelty and advance is their direct demonstration of functional perturbations from the ablation experiments, which is consistent with an inspiratory premotor function of some of the Dbx1-derived IRt neurons.

2) The definitive proof that the inspiratory neurons are XII premotor neurons is to directly demonstrate structurally and functionally synaptic connections with XII inspiratory neurons, which would require more advanced circuit reconstruction techniques than those applied in this study. The authors need to acknowledge these limitations by stating that selectively ablating of identified Dbx1-derived inspiratory neurons with demonstrated synaptic connections to XII motor neurons will be required to more thoroughly establish that the identified subset of Dbx1-derived IRt cells are in fact inspiratory premotor neurons.

Related to this issue, it is not clear why the anatomical reconstructions of loaded neurons shown on panel G, Figure 2, do not show projections towards the XII nucleus for all neurons. The authors should clearly state what are the criteria required to define them as XII premotor neurons.

3) They authors need to acknowledge that they have not determined the transmitter-content of the ablated cells. Thus, while the inspiratory drive to the XII is primarily glutamatergic, it is not exclusively glutamatergic. There is significant GABAergic innervation, see e.g. Jaismal et al. 2015. Similarly, the work by Albert Berger identified the Nucleus of Roller as a major source of inhibitory drive to the XII motoneurons. This is an issue since Dbx1 neurons can also be inhibitory (as stated here, "only 57% are glutamatergic"). Since the authors did not demonstrate that the Dbx1 premotor neurons generate EPSPs in the XII motor neurons, it could well be that among these premotor neurons there are some that are inhibitory. Thus, the authors were not able to address whether Dbx1 IRt premotor neurons are the sole source of glutamatergic inspiratory drive to XII motor neurons, nor could they address whether the Dbx1 IRt premotor neurons also contribute to an inhibitory drive to XII motor drive. The lesion experiment suggests that they contribute to the excitatory drive, but they don't exclude the possibility that some of the lesioned neurons were inhibitory. This issue needs to be stated explicitly.

4) The potential role of contralateral connections is neglected by the authors. They show themselves that some hypothetical XII premotoneurons send axon to the contralateral premotor region (Figure 2) and it is known that some pre-Bötzinger neurons send their axons directly to XII nucleus (Koizumi et al., 2013). The authors should address this point at least in the Discussion.

---

## [Author Response]

Essential revisions:

*1) It is hard to see why is it a significant advance for understanding respiratory circuit function per se to know that a subset of IRt XII inspiratory premotor neurons are Dbx1-derived. Or that Dbx1-expressing progenitors give rise to both rhythm and pattern generating neurons. Knowing the genetic specificity of these cells does not give insight into how genetically defined neuronal populations determine function and connectivity. Rather the advance and novelty of the present study with respect to the previous study is the functional demonstration of a group of cells that are premotor (amplitude modulation not affecting frequency) different from the rhythm generation kernel (frequency and amplitude modulation). The authors should state that the main novelty and advance is their direct demonstration of functional perturbations from the ablation experiments, which is consistent with an inspiratory premotor function of some of the Dbx1-derived IRt neurons.*

We agree with the reviewers that the main contributions of this study with respect to the last study are the functional demonstration that the ablation of IRt Dbx1 neurons disrupts amplitude and not rhythm. These data, combined with the demonstration that a subpopulation of IRt Dbx1 inspiratory modulated neurons appear to be premotoneurons underlie support the key conclusion that Dbx1-derived IRt neurons serve an inspiratory premotor function. We now state this explicitly:

“Therefore, this study advances our previous work (Hayes et al., 2012; Wang et al., 2013; Wang et al., 2014) by demonstrating that Dbx1 IRt neuron ablation disrupts inspiratory burst amplitude but not rhythm. These data strongly suggest that a subpopulation of Dbx1-derived IRt neurons serve an important inspiratory premotor function.”

*2) The definitive proof that the inspiratory neurons are XII premotor neurons is to directly demonstrate structurally and functionally synaptic connections with XII inspiratory neurons, which would require more advanced circuit reconstruction techniques than those applied in this study. The authors need to acknowledge these limitations by stating that selectively ablating of identified Dbx1-derived inspiratory neurons with demonstrated synaptic connections to XII motor neurons will be required to more thoroughly establish that the identified subset of Dbx1-derived IRt cells are in fact inspiratory premotor neurons.*

Two limitations are mentioned here. The first is that we have not provided definitive proof that Dbx1 IRt neurons are XII premotoneurons. The second limitation is that our ablation studies eliminate Dbx1 neurons in the IRt, regardless of whether they are inspiratory or not. The ideal experiment would be to eliminate only those Dbx1 IRt neurons that are also inspiratory-modulated *and* confirmed XII premotoneurons. For clarity, we have addressed each of these limitations separately.

In the first limitation, definitive proof that Dbx1 IRt neurons are XII premotoneurons would require that we directly demonstrate structurally and/or functionally that there are synaptic connections between the putative premotoneuron and XII inspiratory neurons. Anatomical evidence could include EM while functional evidence would require simultaneous dual whole-cell recording from a Dbx1 IRt inspiratory neuron and an inspiratory XII MN that are synaptically connected, and then evoking synaptic potentials in the XII MN. This is not a trivial experiment, which is why none of the papers from the existing literature defining inspiratory XII premotoneurons have provided this evidence either. As in our study, the strongest evidence provided in these previously published papers is antidromic activation from the XII nucleus. To clarify this limitation, we included in the Discussion the statement that:

“While we did not perform dual intracellular recordings to unequivocally show synaptic contact between Dbx1 IRt neurons and XII motoneurons, our conclusion that Dbx1 IRt neurons are XII inspiratory premotoneurons is based on: i) synaptic drive potentials and action potential volleys in Dbx1 IRt neurons that were in phase with inspiratory rhythm; ii) evidence from antidromic activation (including positive collision tests) and axon reconstructions showing that inspiratory Dbx1 IRt neurons project to the XII nucleus; and, iii) the sequential appearance, on average, of inspiratory-related activity first in preBötC neurons, then Dbx1 IRt neurons and finally in the XII nerve.”

The second limitation raised here is that our ablation studies eliminate Dbx1 neurons in the IRt, regardless of whether they are inspiratory or not. A more ideal experiment would be to ablate only those Dbx1 IRt neurons that are also inspiratory-modulated AND confirmed XII premotoneurons. We agree with the reviewers that this would be a much more advantageous experiment. Our automated detection and laser ablation system should – in theory – be able to detect targets on the basis of genetically encoded fluorophores and activity-dependent calcium indicator dyes. However, this has proved very tricky and still cannot be done reliably. Thus, we acknowledge this limitation in the revised manuscript by defining the conditions that should be met for the ideal experiment. In the Discussion, we now state:

“However, there are two main limitations to the ablations experiments. First, Dbx1 IRt neurons detected for ablation included inspiratory and non-inspiratory neurons. […] We predict a more precipitous, and possibly complete, decline in inspiratory burst amplitude if bona fide Dbx1 IRt inspiratory premotoneurons could be selectively ablated, avoiding Dbx1 IRt neurons with alternative functions.”

Related to this issue, it is not clear why the anatomical reconstructions of loaded neurons shown on panel G, Figure 2, do not show projections towards the XII nucleus for all neurons.

Figure 2 is the population of neurons that had anatomically reconstructed contralateral projections. These Dbx1 inspiratory-modulated neurons were in the IRt and thus considered putative preMNs. A subset of these neurons (i.e. on the left-hand side of the slice) was also antidromically activated. There are a number of reasons why any anatomical reconstruction does not show projections to XII, including: 1) axon collaterals to the XII nucleus may be superficial and then severed during the slicing process even while commissural axon collaterals are evident, 2) axons may course deeply into the slice and thus are too deep in tissue to image, and 3) neuron fills and reconstructions are not always successful.

*The authors should clearly state what are the criteria required to define them as XII premotor neurons.*

We have clarified the criteria in the text for defining neurons in the present study as XII inspiratory premotoneurons (Results):

“Criteria for inclusion in our putative premotoneuron sample were somatic location in the IRt, tdTomato expression, and inspiratory modulation. Additional evidence obtained in some cells included antidromic activation from the XII nucleus, positive collision tests, and axonal projections to the ipsilateral XII nucleus.”

We have also removed three neurons from the reported database. These three neurons were all antidromically activated from the XII nucleus so very likely premotoneurons. Anatomically, however, they sit at the dorsal border of the preBötC. They can be seen in Figure 2 of the previous version; they are the three the ventral-most cells. Two of these were also in Figure 2 where they were shown as contralaterally projecting neurons with cell bodies on the left side of the slice. These cells have been removed from all figures, the numbers of recorded neurons were changed accordingly, and the electrophysiological properties reported in association with Figure 3 have also been recalculated.

We removed these neurons to clarify the criteria for inclusion as a putative premotoneuron. As an Advance article, the novel aspect of this study compared to our previous work is that it now reports the properties of Dbx1 IRt neurons, and the effects of ablating Dbx1 IRt neurons on inspiratory modulated XII output. The three neurons that have been removed from the study were on the preBötC side of the border between IRt and preBötC. Removing these three neurons does not change any of the conclusions, but now all cells included in the study are unambiguously in the IRt.

*3) They authors need to acknowledge that they have not determined the transmitter-content of the ablated cells. Thus, while the inspiratory drive to the XII is primarily glutamatergic, it is not exclusively glutamatergic. There is significant GABAergic innervation, see e.g. Jaismal et al. 2015. Similarly, the work by Albert Berger identified the Nucleus of Roller as a major source of inhibitory drive to the XII motoneurons. This is an issue since Dbx1 neurons can also be inhibitory (as stated here, "only 57% are glutamatergic"). Since the authors did not demonstrate that the Dbx1 premotor neurons generate EPSPs in the XII motor neurons, it could well be that among these premotor neurons there are some that are inhibitory.*

We agree completely that a portion of the ablated IRt Dbx1 neurons are likely to be GABAergic. We now discuss this in detail and make the point that this will not significantly impact our conclusion that a subpopulation of Dbx1 IRt neurons are glutamatergic and have a significant excitatory inspiratory premotor function. We now state in the Discussion that:

“A second limitation is that the ablated Dbx1 IRt neurons have unknown transmitter phenotype. Just over half of Dbx1 IRt neurons are glutamatergic. […] Thus, in the unlikely circumstance that we ablated endogenously active inhibitory Dbx1 IRt premotoneurons, it would underestimate rather than contradict our conclusion that Dbx1 IRt premotoneurons provide excitatory drive for inspiratory motor output.”

Thus, the authors were not able to address whether Dbx1 IRt premotor neurons are the sole source of glutamatergic inspiratory drive to XII motor neurons […]

We agree completely so state in the Discussion:

“Whether they are the sole source is uncertain. Persistence of inspiratory XII nerve activity following laser ablation of Dbx1 IRt neurons might suggest additional non-Dbx1 derived sources of inspiratory premotor drive. However, the ablation-resistant activity could also derive from Dbx1 premotoneurons too deep for laser ablation, or Dbx1-derived neurons that went untargeted due to lack of tdTomato expression (see Methods).”

*[…] Nor could they address whether the Dbx1 IRt premotor neurons also contribute to an inhibitory drive to XII motor drive. The lesion experiment suggests that they contribute to the excitatory drive, but they don't exclude the possibility that some of the lesioned neurons were inhibitory. This issue needs to be stated explicitly.*

This issue has been discussed in detail as described above in our first response to comment 3.

*4) The potential role of contralateral connections is neglected by the authors. They show themselves that some hypothetical XII premotoneurons send axon to the contralateral premotor region (Figure 2) and it is known that some pre-Bötzinger neurons send their axons directly to XII nucleus (Koizumi et al., 2013). The authors should address this point at least in the Discussion.*

We agree with the reviewers that the potential role of contralateral projections is very interesting. However, we are unclear on how to further elaborate on the potential significance of contralateral projections. We feel we have already covered the most likely possibilities when we state (Discussion):

“Contralateral projections from Dbx1 IRt inspiratory putative premotoneurons were unexpected and of unknown function. They may contribute to the contralateral inspiratory drive for airway and tongue movements (Onoet al., 1994; Peeveret al., 2002), they may be inspiratory rhythmogenic preBötC neurons displaced into the IRt, or they may help coordinate inspiration with other oromotor behaviours (Onoet al., 1998; Tupalet al., 2014)”

We are not clear on what other potential roles these contralateral projections might play.